# Quality of Life in Vulnerable Older Patients with Metastatic Colorectal Cancer Receiving Palliative Chemotherapy—The Randomized NORDIC9-Study

**DOI:** 10.3390/cancers13112604

**Published:** 2021-05-26

**Authors:** Gabor Liposits, Henrik Rode Eshøj, Sören Möller, Stine Brændegaard Winther, Halla Skuladottir, Jesper Ryg, Eva Hofsli, Carl-Henrik Shah, Laurids Østergaard Poulsen, Åke Berglund, Camilla Qvortrup, Pia Österlund, Bengt Glimelius, Halfdan Sorbye, Per Pfeiffer

**Affiliations:** 1Department of Oncology, Odense University Hospital, 5000 Odense, Denmark; Stine.Winther@rsyd.dk (S.B.W.); per.pfeiffer@rsyd.dk (P.P.); 2Department of Clinical Research, University of Southern Denmark, 5000 Odense, Denmark; Soren.Moller@rsyd.dk (S.M.); Jesper.Ryg@rsyd.dk (J.R.); 3Academy of Geriatric Cancer Research (AgeCare), 5000 Odense, Denmark; camilla.qvortrup@regionh.dk; 4Quality of Life Research Center, Department of Haematology, Odense University Hospital, 5000 Odense, Denmark; henrik.eshoj@rsyd.dk; 5OPEN-Open Patient Data Explorative Network, Odense University Hospital, 5000 Odense, Denmark; 6Department of Oncology, Regional Hospital West Jutland, 7400 Herning, Denmark; hallskul@rm.dk; 7Department of Geriatric Medicine, Odense University Hospital, 5000 Odense, Denmark; 8Department of Oncology, Trondheim University Hospital, 7030 Trondheim, Norway; eva.hofsli@ntnu.no; 9Department of Clinical and Molecular Medicine, Norwegian University of Science and Technology, 7030 Trondheim, Norway; 10Theme Cancer, Karolinska University Hospital, 171 76 Stockholm, Sweden; carl-henrik.shah@sll.se; 11Department of Oncology, Aalborg University Hospital, 9000 Aalborg, Denmark; laop@rn.dk; 12Department of Immunology, Genetics and Pathology, Uppsala University, 751 05 Uppsala, Sweden; ake.berglund@igp.uu.se (Å.B.); bengt.glimelius@igp.uu.se (B.G.); 13Department of Oncology, Tampere University Hospital and Tampere University, 33100 Tampere, Finland; pia.osterlund@hus.fi; 14Karolinska Institute and Karolinska University Hospital, 171 76 Stockholm, Sweden; 15Department of Oncology, Helsinki University Hospital, 00014 Helsinki, Finland; 16Department of Oncology, Haukeland University Hospital, 5021 Bergen, Norway; halfdan.sorbye@helse-bergen.no; 17Department of Clinical Science, University of Bergen, 5021 Bergen, Norway

**Keywords:** older adults, metastatic colorectal cancer, chemotherapy, vulnerability, quality of life, physical functioning, EORTC QLQ-C30, geriatric oncology

## Abstract

**Simple Summary:**

Bowel cancer is one of the leading cancer-types in both sexes worldwide. Despite that most new cases and deaths occur in people aged 70 years or older, few clinical trials have investigated the best way to administer chemotherapy in older or frail patients. The NORDIC9-study established that moderately dose-reduced combination chemotherapy improved survival without extra side-effects compared to full dose single drug therapy. However, many older patients with incurable cancer seem to prefer preserved quality of life rather than longer survival. Therefore, our aim with the current quality of life analysis of the NORDIC9-study was to assess that the more effective chemotherapy was not at the expense of decreased quality of life. Our analyses showed that moderately dose-reduced combination chemotherapy-maintained quality of life, physical functioning, and resulted in less symptoms than treatment with full dose single drug in older patients not tolerating standard combination chemotherapy usually provided to young and fit patients.

**Abstract:**

Quality of life data from randomized trials are lacking in older patients with metastatic colorectal cancer (mCRC). In the randomized NORDIC9-study, reduced-dose S1+oxaliplatin (SOx) showed superior efficacy compared to full-dose S1 monotherapy. We hypothesized that treatment with SOx does not result in inferior quality of life. Patients with mCRC aged ≥70 years and that were *not* a candidate for standard combination chemotherapy were included and randomly assigned to receive either S1 or SOx. The EORTC QLQ-C30 questionnaire was completed at baseline, after 9, and 18 weeks. The primary endpoint was global Quality of Life (QoL) at 9 weeks. For statistical analysis, a non-inferiority design was chosen applying linear mixed effects models for repeated measurements. The results were interpreted according to statistical significance and anchor-based, clinically relevant between-group minimally important differences (MID). A total of 160 patients aged (median (Interquartile range (IQR))) 78 years (76–81) were included. The QLQ-C30 questionnaire was completed by 150, 100, and 60 patients at baseline, at 9, and 18 weeks, respectively. The difference at 9 weeks in global QoL was 6.85 (95%CI—1.94; 15.65) and 7.37 (0.70; 14.05) in the physical functioning domain in favor of SOx exceeding the threshold for MID. At 18 weeks, the between-group MID in physical functioning was preserved. Dose-reduced combination chemotherapy may be recommended in vulnerable older patients with mCRC, rather than full-dose monotherapy.

## 1. Introduction

Colorectal cancer (CRC) predominantly affects older adults [1]; 56% of incidence and 68% of mortality occur in patients aged ≥65 years [2]. In the older population [3], comorbidities, impaired organ function, and geriatric-syndromes (e.g., fall-tendency, osteoporosis, urinary incontinence) are frequently present resulting in decreased physiological reserves, called vulnerability [4,5]. Vulnerability affects treatment outcomes due to compromising standard treatment approach [6], thus, in the case of chemotherapy, monotherapy and/or dose-reduction are commonly applied. The optimal approach regarding palliative systemic treatment in older vulnerable patients with metastatic colorectal cancer (mCRC) remains unclear. The randomized NORDIC9-study explored this by including patients with mCRC aged ≥70 years and that were *not* a candidate for standard treatment, by comparing reduced-dose combination chemotherapy with S1+oxaliplatin (SOx) or full-dose monotherapy (S1) (Teysuno, Taiho Pharmaceutical Co. Ltd., Tokyo, Japan). The primary results established that SOx resulted in significantly improved progression-free survival (PFS) (SOx; 6.2 vs S1; 5.3 months, hazard ratio: 0.72 (95%CI: 0.52–0.99), *p* = 0.047). Additionally, patients treated with SOx experienced less toxicity, and fewer hospitalizations, compared to S1 [7].

When the NORDIC9-study was designed [8], we aimed for prolonged PFS with reduced-dose combination therapy, though somewhat higher risk of adverse events due to the combination of cancer drugs was predicted. Therefore, we also evaluated patient-reported health-related quality of life (QoL) and physician-reported functioning indicated as Eastern Cooperative Oncology Group Performance Status (ECOG PS).

Only little is known about QoL, functional status, and symptom burden in vulnerable older patients with mCRC receiving palliative chemotherapy. Oncologists are usually interested in traditional tumor-centered endpoints used in randomized controlled trials (RCTs) of cancer drugs [9,10,11]. Overall survival (OS) is considered as the gold standard primary endpoint of efficacy in RCTs [9]. When several treatment lines are available, measuring OS may be challenging, therefore, surrogate outcomes for OS like progression-free survival and response rates have been widely used. Whether these surrogate outcomes are valid for OS or relevant markers for expanded length of survival or QoL, is questionable. Undoubtedly, if a treatment results in prolonged OS and less toxicities, thus, showing clear benefits, measuring QoL from a patient-perspective may be less meaningful [12,13]. When two treatment options gain the same OS benefit, though, with one of them having more adverse events, choosing the better option is easy for the physicians and patients. In contrast, when the PFS benefit only slightly differs between treatment arms or between active treatment and placebo/best supportive care, with differences in toxicities, measuring QoL is important [14]. In this case, patient-centered QoL data may add important information when patients and caregivers are involved in shared decision-making. Furthermore, an important aspect of palliative treatment is that QoL tends to worsen across treatment lines due to disease progression, increasing tumor burden, and cumulative toxicities [15]; therefore, QoL becomes increasingly important through the continuum of care.

Despite that incidence and mortality rates top in adults aged ≥70 years, older patients with cancer are underrepresented in RCTs [16,17,18,19,20]. The older population is generally treated based on extrapolated data from highly selected younger and healthier cohorts; however, the same drugs may be less effective and more toxic in older individuals having comorbidities and impaired organ function. In addition, many older patients with cancer tend to prioritize improvement or preservation of QoL [21,22,23], rather than prolonged survival.

RCTs addressing QoL and patient-centered endpoints have long been desired by patients [24,25], by researchers and physicians [26,27,28,29,30], the leading cancer societies [31,32] and by regulatory authorities [19,33]. However, QoL or other patient-centered endpoints are either not included in RCTs of cancer drugs at all [10,11], or if they are, the results are often insufficiently reported due to methodological and practical issues [34]. In a systematic review of phase III CRC trials conducted between 2012 and 2018, Lombardi and colleagues found that 61% of the trials did not include QoL as endpoint [10]. In 39% of the trials including QoL among endpoints in their primary publication, QoL data were not yet reported. In RCTs investigating treatment options in the metastatic setting, where attention to QoL is of particular interest due to limited life expectancy, QoL data were not available in two-thirds of trials. This stresses the importance of appropriate conducting and reporting of RCTs regarding patient-centered endpoints.

The overarching objective of the present study was to investigate whether prolonged PFS with reduced-dose combination chemotherapy comes at the expense of poorer QoL and physical functioning. We hypothesized that treatment with reduced-dose SOx might result in inferior QoL and physical functioning compared to treatment with full-dose S1 monotherapy.

## 2. Materials and Methods

### 2.1. Study Design, Participants, and Interventions

The NORDIC9-study was a prospective, randomized open-label phase II multi-center study including older vulnerable patients aged ≥70 years with mCRC that were *not* candidates for standard combination chemotherapy. Participants were recruited from 23 centers in four Nordic countries. Detailed protocol, study design including CONSORT flow diagram, findings of the primary endpoint, and several secondary endpoints have been published [7,8]. National ethical committees in all countries approved the study, which was conducted according to the Declaration of Helsinki and Good Clinical Practice guidelines. The NORDIC9-trial was registered at EudraCT (reg. no. 2014-000394-39).

The current manuscript conforms to the CONSORT 2010 statement for reporting RCTs with patient-reported outcomes as primary or secondary outcomes [35,36]. A statistical analysis plan, including pre-specified thresholds for clinically relevant between-group minimal important differences (MID), was prepared prior to analyses and made available online (https://portal.findresearcher.sdu.dk/da/publications/statistical-analysis-plan-for-nordic9-study-quality-of-life-analy, accessed on 9 March 2021), thus, securing transparency and reproducibility.

### 2.2. Interventions

Patients were randomly assigned (allocation ratio 1:1) to receive either reduced-dose SOx (S1 20 mg/m^2^ orally twice daily on days 1–14 + oxaliplatin 100 mg/m^2^ intravenously on day 1, q3w (once every 3 weeks)) or full-dose S1 (30 mg/m^2^ orally twice daily on days 1–14, q3w). Dose modification was allowed according to the treatment protocol. The addition of bevacizumab (7.5 mg/kg intravenously, q3w) was optional at the discretion of the treating physician, but the decision was required before randomization. Evaluation of response was planned after every three cycles. Patients were treated until disease progression/death or unacceptable toxicities, whichever occurred first.

### 2.3. Outcomes

#### 2.3.1. Patient-Reported Outcomes

Patient-reported outcomes (PRO) on QoL were assessed with the European Organization for Research and Treatment of Cancer Quality of Life Questionnaire-Core 30 version 3.0 (EORTC QLQ-C30) [37]. The EORTC QLQ-C30 was chosen as it is widely applied in QoL research and extensively used in patients with solid cancer. The EORTC QLQ-C30 contains a global health scale, five functioning scales (physical, role, cognitive, emotional, and social functioning), three symptom scales (fatigue, nausea and vomiting, and pain), and six single items (appetite loss, diarrhea, dyspnea, constipation, insomnia, and financial impact). The questionnaire asks about the last one-week time frame at completion and uses a four-point response format (“not at all”, “a little”, “quite a bit”, and “very much”). The only exception is the global health scale using a seven-point response format ranging from “very poor” to “excellent”. The different scale and item scores are linearly transformed to a score between 0 and 100 according to the scoring manual [38]. For global QoL and the functioning scales, a higher score indicates better outcomes, for symptom scales, a higher score reveals more prominent symptoms.

#### 2.3.2. Physician-Reported Physical Functioning

Physician-reported outcomes of physical functioning were assessed through the ECOG PS. The ECOG PS score has six possible values where 0 means that the patient is “fully active, able to carry on all pre-disease performance without restriction” and a score of 5 means that the patient is “dead”. The ECOG PS is the most frequently used clinical score in oncology daily practice and research.

### 2.4. Data Administration and Timing of Assessments

Patient-reported and physician-reported data were collected at the time of randomization (baseline), after 9-, and 18-week follow-ups, corresponding to the administration of three and six cycles of chemotherapy. For the EORTC QLQ-C30, patients self-completed the questionnaire in paper-based versions where after it was collected by qualified study nurses and stored in patient documentation at the clinical research units. For the ECOG PS, physicians noted the score at the time of the clinical visit and stored it in the participants’ electronic medical record.

### 2.5. Data Compliance

Compliance of the patient-reported outcomes was calculated using the proportion of randomized patients with completed questionnaires and the proportion of patients expected to complete questionnaires (alive and still on study) at all scheduled visits (at 9-week and 18-week follow-ups).

### 2.6. Endpoints

PRO objectives, hypotheses, and analyses were pre-specified in a statistical analysis plan prior to data analyses.

#### 2.6.1. Primary Endpoint

The primary endpoint of the current analyses was to test the hypothesis that treatment with reduced-dose SOx is not inferior to full-dose S1 in terms of affecting global QoL. We did this by comparing the impact of SOx vs. S1 on changes in the global QoL domain as assessed by the questionnaire from randomization to the 9-week follow-up. We choose to compare the 9-week follow-up scores to baseline, because in patients with considerable tumor load, the first two months are essential for whether an objective response or disease stabilization is obtained or not. Moreover, toxicity generally occurs at the initiation of the treatment, thus, both factors may alter global health.

While recognizing that the value of using the global QoL domain as primary outcome in clinical trials investigating mCRC has been questioned due its low sensitivity to changes [39], the effect of palliative chemotherapy on global QoL has not yet been extensively studied in older vulnerable patients with cancer. We strongly believed this to be an important patient-centered outcome in the NORDIC9-study; the global QoL domain reflects the patients’ self-perceived general health and well-being, and these aspects are often prioritized by older patients with incurable cancer and limited life expectancy, rather than prolonged OS [21,22,23].

#### 2.6.2. Secondary Endpoints

##### Key Secondary Objectives

This included explorative analyses of the effects of SOx relative to S1 on other clinically important patient-reported outcome measurements in the following EORTC QLQ-C30 subscales: (1) physical functioning, (2) role functioning, and (3) social functioning from baseline to the 9-, and 18-week follow-ups.

We selected domains of functioning as these were considered to play a key role in older patients regarding the maintenance of their general physical and mental well-being. We hypothesized that SOx would not result in inferior outcomes compared to S1.

##### Other Secondary Objectives

Given the toxicity profile of the applied cytotoxic agents and the adverse events reported in the primary publication of the NORDIC9-study [7], we also aimed to perform hypothesis generating explorative analyses on the difference between the two treatment arms from baseline to 9-, and 18-week in the following EORTC QLQ-C30 symptom domains and single items: (1) fatigue, (2) dyspnea, (3) nausea and vomiting, (4) diarrhea, and (5) pain.

We hypothesized that patients receiving S1 full-dose monotherapy would experience lower toxicity but may be higher symptom burden due to less efficacy compared to SOx.

Furthermore, the risk of deterioration in physician-reported physical functioning specified by the ECOG PS score at the 9-week follow-up was established. We hypothesized that the risk of deterioration was higher in patients receiving SOx compared to S1.

### 2.7. Statistical Consideration and Analysis Metrics

Formal sample size calculation for the PRO data was not performed. Hence, the current sample follows the sample size needed for the primary endpoint of the NORDIC9-study [7]. The current analyses of patient-centered outcomes were based on all randomized patients receiving ≥1 dose of chemotherapy and completing at least one QoL assessment (*n* = 150), while the physician-reported outcome, risk of deterioration based on ECOG PS was conducted on the intention to treat (ITT) population of the randomized trial (*n* = 160).

#### 2.7.1. For the Patient-Reported Outcomes

The primary statistical model for the PROs consisted of linear mixed effects model for repeated measures (MMRM) including all PRO assessments (e.g., baseline, 9 weeks, and 18 weeks). This allowed us to compare the average treatment effect of reduced-dose SOx vs. full-dose S1.

These analyses were performed as non-inferiority analyses reporting coefficients with 95% confidence intervals for mean differences and comparing with between-group mean minimal important differences (MID) as reported by Musoro et al. [40]. As it is not assured that the baseline PRO measurements were performed before the patient and healthcare professionals were informed about the result of the randomization, the treatment effect was included at baseline. Methodological discussion of this consideration can be found elsewhere [41].

To provide easily interpretable estimates of the probabilities of non-inferiority and superiority we provided a sensitivity analysis fitted a Bayesian linear mixed effects model (using Markov Chain Monte Carlo, MCMC) with weakly informative symmetric priors (N (0,10000)) for regression coefficients and weakly informative (IGamma(0.01,0.01)) priors for variance components. This model resulted in posterior probabilities of superiority, non-inferiority, equivalence, and inferiority, providing straightforward comparison between possible scenarios of treatment effects among arms.

#### 2.7.2. For the Physician-Reported Outcomes

Deterioration of ECOG PS score from baseline to 9-, and 18-week follow-ups was defined as a ≥1-point decline from baseline and was analyzed using logistic regression. In the main analysis, discontinuation of the study participation for any reason was regarded as deterioration of performance. Two sensitivity analyses (A and B) were conducted. In sensitivity analysis “A” only discontinuation due to death was considered as deterioration, while patients who left the study for other reasons were excluded. In sensitivity analysis “B” only observed ECOG PS were used to define deterioration and any patients who left the study were excluded.

### 2.8. Data Analysis

All data were analyzed in STATA v16 (StataCorp LLC, College Station, TX, USA), *p*-values ≤ 0.05 were considered statistically significant and estimates were reported with 95% confidence intervals (95%CI).

### 2.9. Adjustment for Multiplicity

As only global QoL was considered as the primary endpoint along with a pre-specified primary data assessment time point, and other outcomes were considered secondary and supportive to the primary outcome, no adjustment for multiplicity was performed.

### 2.10. Missing Data, Sensitivity Analyses, and Robustness

Missing data in the PRO analyses were handled by restricted maximum likelihood estimation in the linear mixed effects model including all answered questionnaires. This analysis assumed missing at random questionnaires, although this assumption could not be checked in our study. Missing data in the ECOG PS analysis were handled by the two pre-specified sensitivity analyses (see above).

Acceptable fulfillment of model assumptions was investigated by normal quantile-quantile plots for residuals and random effects in the linear mixed models, in case of deviations, nonparametric bootstrapping with 1000 repetitions was performed.

Performance of the MCMC algorithm applied in the Bayesian models was investigated by diagnostic plots (trace, autocorrelation, and posterior kernel density).

Goodness of fit of the logistic regression model was investigated by Hosmer–Lemeshow’s test.

### 2.11. Interpretation

In general, each PRO domain was interpreted according to clinically relevant MID using guidelines for between-group differences and changes over time [40,42].

#### 2.11.1. Primary Outcome

For the global QoL domain, the result was interpreted according to the anchor-based MID for between-group differences as reported by Musoro et al. [40], who established MIDs specifically in patients with mCRC. Based on the principles related to non-inferiority designs we pre-specified that a 95% CI excluding negative differences between groups of greater than −8.13 points (the mean MID for deterioration) in global QoL was interpreted as indicating the absence of a clinically meaningful inferiority difference.

#### 2.11.2. Key Secondary Patient-Reported Outcomes

For functioning domains (physical functioning, role functioning, and social functioning), symptom domains, and single items (fatigue, nausea and vomiting, diarrhea, and pain) the results were interpreted according to the anchor-based mean MID for deterioration by between-group differences according to Musoro et al. [40]. However, as the symptom domain of dyspnea was not included in [40], this domain was interpreted according to the medium between-group MID score (at least 9 points difference) established by Cocks et al. [42].

#### 2.11.3. Clinician-Reported Outcome

Risk of deterioration was compared between the two treatment groups according to the analysis specified above.

## 3. Results

### 3.1. Patient Population

From March 2015 to October 2017, 160 patients with a median age of 78 years (interquartile range 76–81) were included in the NORDIC9-study, of whom 150 were available for the QoL analysis (Figure 1). Patient and disease characteristics and baseline score of QLQ-C30 domain scores were well balanced between the treatment arms as shown in Table 1 and Table 2.

### 3.2. Compliance

The EORTC QLQ-C30 questionnaire was completed by all 150 (100%) patients at baseline. Patients alive and still on-study at 9-week and 18-week follow-ups, thus constituting the proportion of number of patients expected to return a questionnaire, were 132 and 114, respectively. Questionnaire compliance at 9 weeks was 76% (100/132) and 52% (60/114) at 18 weeks. Compliance was balanced between groups at all assessment time points.

Most missing questionnaires were lacking due to a procedural error; one site including 36 patients completed EORTC QLQ-C30 only at baseline.

### 3.3. Primary Endpoint

The between-group difference in global QoL was 6.85 (95%CI -1.94; 15.65) at 9 weeks in favor of SOx (SOx: 2.63 (−4.84; 10.11), S1: −4.22 (−9.16; 0.72), *p* = 0.127). Despite this being non-significant, the difference exceeded the pre-specified threshold for MID between groups. At 18 weeks, the difference was neither statistically significant nor clinically relevant (Table 3, Figure 2).

Regarding dyspnea, clinically-relevant MID was defined according to the EORTC evidence-based guidelines for between-group differences by Cocks et al.

### 3.4. Key Secondary Patient-Reported Endpoints

In the physical functioning domain, a statistically significant between-group difference of 7.37 (0.70; 14.05) at 9 weeks was seen in favor of SOx (SOx: −0.14 (−4.73; 4.44), S1: −7.52 (−12.23; −2.80), *p* = 0.03). This difference exceeded the pre-specified threshold for between-group MID. At 18 weeks, a non-significant, however clinically relevant between-group difference of 7.88 (−1.68; 17.43) was seen in favor of the SOx group in physical functioning (SOx: −2.57 (−9.50; 4.36), S1: −10.44 (−16.98; −3.91), *p* = 0.106).

Regarding the domains of social and role functioning, neither statistically significant differences, nor clinically relevant MIDs were observed at 9 and 18 weeks (Table 3).

### 3.5. Other Secondary Endpoints

Regarding the EORTC QLQ-C30 symptom domains and single items, statistically significant between-group difference of −11.36 (−21.18; −1.53) was seen in dyspnea at 9 weeks in favor of the SOx group, which also exceeded the pre-specified threshold for between-group MID (SOx: −3.08 (−10.18; 4.01), S1: 8.27 (1.70; 14.85), *p* = 0.024). The between-group difference slightly decreased at 18 weeks, though the trend was not altered (Table 3).

In fatigue, no statistically significant between-group difference was observed, the SOx group experienced less fatigue, though, the between-group MID did not exceed the threshold for clinical relevance at 9 weeks. At 18 weeks, clinically relevant between-group difference of -7.54 (−20.09; 5.00) was observed in favor of SOx (SOx: 1.24 (−7.93; 10.40), S1: 8.78 (0.21; 17.36), *p* = 0.239) (Table 3).

In terms of diarrhea, nausea and vomiting, and pain, neither statistically significant differences nor clinically relevant MID were observed during follow-up (Table 3).

### 3.6. Clinician-Reported Outcome

From baseline to 9 weeks of treatment, the risk of deterioration in ECOG PS was lower in the SOx group compared to the S1 group. However, the result was not statistically significant (Odds ratio (OR) = 0.75 (95% CI 0.40; 1.41), *p* = 0.366). Similar result was observed at 18 weeks of treatment (OR = 0.88 (0.47; 1.64), *p* = 0.676) (Table 4). The results of the sensitivity analyses are shown in Table 5.

### 3.7. Bayesian Sensitivity Analyses

The sensitivity analyses resulted in a probability of non-inferiority of the reduced-dose combination chemotherapy of at least 89% for all investigated EORTC QLQ-C30 domains at 9 weeks, and at least 76% at 18 weeks. The only deviation from this was the fatigue domain, for which the evidence of non-inferiority was only 64% and 46% at 9 and 18 weeks, respectively (Appendix A). The posterior distribution for global QoL at 9 weeks is presented in Appendix A.

## 4. Discussion

### 4.1. Summary of Findings

Our findings demonstrate that reduced-dose combination chemotherapy may be the treatment of choice in older vulnerable patients with mCRC when preservation of QoL, physical functioning, and symptom relief are important goals of the treatment. Although non-significant, a clinically relevant between-group MID in favor of SOx was observed in the primary endpoint, global QoL at the primary assessment time point at 9 weeks after commencing the treatment. Additionally, a statistically significant and clinically relevant difference in the key secondary endpoint of physical functioning in favor of SOx treatment was seen at the primary data assessment time point. Likewise, our findings indicate that patients treated with SOx experienced less symptom burden with clinically relevant between-group MID in symptoms of dyspnea and fatigue at 9 weeks and at 18 weeks, respectively. The Bayesian sensitivity analyses confirmed our findings of good evidence of non-inferiority of the reduced-dose combination chemotherapy.

### 4.2. Explanations/Interpretations

Patients receiving reduced-dose SOx gained PFS benefit, experienced less toxicity [7], and preserved global QoL and physical functioning compared to those treated with full-dose S1 monotherapy. The difference in terms of global QoL occurred mostly due to the deterioration in the S1 group at 9 weeks, while global QoL scores in the SOx group remained stable during treatment. Furthermore, the S1 group experienced more prominent symptoms most probably due to toxicity and less efficacy.

### 4.3. Comparison to Other Studies

Although the global QoL seems to provide beneficial information, its value and sensitivity reflecting to the toxicity issues patients are facing during palliative treatment for mCRC were questioned by Schuurhuizen and colleagues [43,44]. In their systematic review, 30 phase III RCTs evaluating palliative systemic treatments in patients with mCRC were assessed based on QoL, toxicities, and primary outcomes [43]. Interestingly, in 83% of these RCTs no difference between treatment arms was observed in terms of global QoL, although, 73% of the studies reported higher occurrence of grade 3–5 *non*-hematological toxicities in the experimental arm. In 86% of these studies, against expectations [43,44], global QoL was either not affected or improved. In 37% of studies where primary outcome was improved, no enhancement in global QoL was observed. A reasonable question was articulated by the authors, whether global QoL is sensitive enough and thus should be considered as a clinically relevant endpoint. In these studies, global QoL score does not seem sensitive enough to guide treatment selection and shared decision-making. This contrasts with our findings in older patients showing a clinically relevant difference, thus, seems able to guide treatment selection. It should be noted that the systematic review contained only one study specifically designed for older patients applying exclusively the global QoL domain from the EORTC QLQ-C30 questionnaire [45]. Furthermore, QoL improvement was defined as “any increase between baseline and 12 weeks”, thus, scoring procedure did not follow the official EORTC manual, and the results must be interpreted with caution. Another explanation is that we applied cancer site specific anchor-based clinically important MID for interpretation of our findings, making the interpretation more specific to the group of patients with mCRC.

Another option might be the application of risk of deterioration or time to deterioration as endpoint, based on clinician-reported outcomes, like ECOG PS; however, ECOG PS and oncologists’ clinical judgement have shown poor association compared to a systematic in-depth evaluation, called comprehensive geriatric assessment (CGA) in older patients with cancer [46,47]. Hence, ECOG PS might not be appropriate to reveal impairments which may be recognized through CGA and do affect QoL. The solution may be to collect data from the older population through randomized clinical trials dedicated and designed to this specific group of patients [26,27,28,29,30]; furthermore, widespread implementation of the CGA for patient stratification is desired [48,49,50].

### 4.4. Methodological Considerations-Strengths and Limitations

A major strength of our study is that the statistical analysis plan was developed and made available online before the analyses were conducted securing transparency and reproducibility. In addition, to the best of our knowledge, our study was the first applying the MID for interpreting the EORTC QLQ-C30 for objective comparison and interpretation of QoL domains in older patients with mCRC treated with chemotherapy [40]. Using MID may allow easier understanding of the potential benefits and harms of the treatment, hence, focusing on PROs and including them in shared decision-making process will thus be more applicable in the clinical practice. Finally, we included vulnerable older patients with mCRC in a multicenter study. This important group of patients has not previously been extensively investigated and we believe our results contribute significantly to this field.

It must be acknowledged, that our dataset has some limitations due to missing items and questionnaires; however, in older vulnerable adults with metastatic cancer the attrition leading to missing data is inevitable considering the advanced cancer and vulnerabilities these patients are presented with. Patients with most symptoms having lowest baseline scores are at highest risk for drop-out, thus, changes in MIDs might be biased at 9 and 18 weeks. It is expected at patient with higher baseline scores or obtaining response will complete the questionnaires during the treatment course, which reflects to the general perception of QoL of patients who were less vulnerable or derived benefit from the treatment. Notably, the number of missing items and questionnaires due to patient drop-out is consistent with previously reported data from a similar population [51]. To deal with missing items and drop-out appropriately, we strictly followed the EORTC scoring manual and carried out sensitivity analyses with different imputation choices for the ECOG secondary outcome allowing utilization of data in incomplete observations. Missing questionnaires due to protocol deviation and procedural errors emphasizes the importance of appropriate conductance of RCTs.

## 5. Conclusions

The NORDIC9-study established that reduced-dose combination chemotherapy in addition to its increased efficacy did not result in inferior QoL compared to the reference treatment with full-dose monotherapy in older patients not fit for standard combination treatment. Thus, the treatment of choice in older vulnerable patients with mCRC should be reduced-dose combination chemotherapy when preservation of QoL, physical functioning, and symptom relief are important goals of the treatment.

## 6. Future Perspectives/Implication

Many older patients with cancer tend to prioritize improvement or preservation of QoL rather than overall survival. Therefore, patient-centered endpoints including patient preferences and shared decision-making with a focus on preservation and improvement of physical and mental wellbeing should be the primary or co-primary endpoints of future clinical trials in older patients with cancer, rather than PFS and surrogate outcomes widely applied in medical oncology. Likewise, for appropriate patient stratification, the concept of CGA should be incorporated and guide treatment and supportive care interventions.

## Figures and Tables

**Figure 1 cancers-13-02604-f001:**
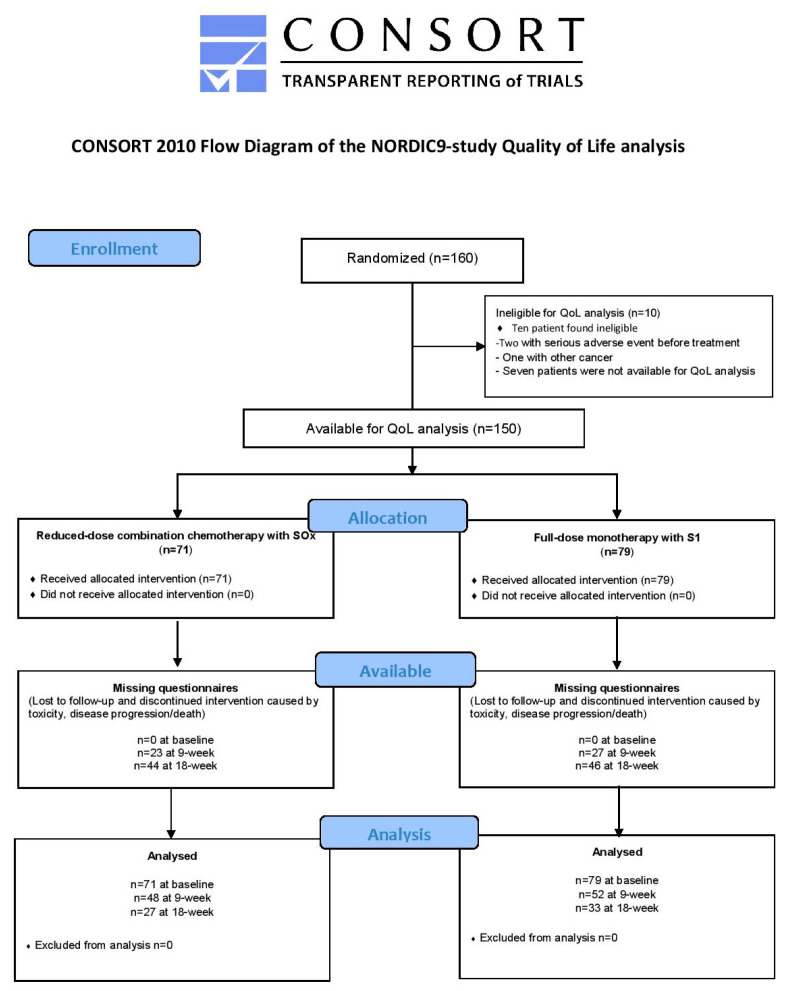
Patients available for quality of life analysis in the randomized NORDIC9-study, reported according to CONSORT flowchart.

**Figure 2 cancers-13-02604-f002:**
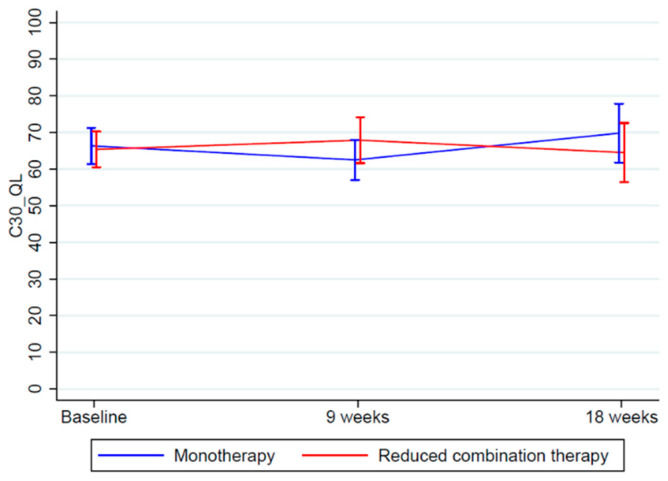
Global QoL over time as line-diagram with 95% CI at baseline, at 9-week and at 18-week follow-ups. Higher values indicate better global QoL.

**Table 1 cancers-13-02604-t001:** Baseline demographic and clinical characteristics of patients included in the quality of life analyses of the randomized NORDIC9-study.

Characteristics Data Presented as Mean or *n* (%) as Appropriate	NORDIC9-Study Treatment Arms Patients Available for Quality of Life Analysis (*n* = 150)	
	Full-Dose Monotherapy Arm A *n* = 79	Reduced-Dose Combination Chemotherapy (CT) Arm B *n* = 71
**Age**		
**Mean age in years (SD)**	78.4 (4.0)	77.8 (3.7)
70–74 years	14 (18%)	17 (24%)
75–79 years	36 (46%)	34 (48%)
≥80 years	29 (37%)	20 (28%)
**Gender**		
Female	39 (49%)	36 (51%)
Male	40 (51%)	35 (49%)
**ECOG Performance status ***		
0	31 (40%)	22 (31%)
1	32 (41%)	35 (49%)
2	15 (19%)	14 (20%)
**Location of primary tumor ***		
Left sided colon	25 (32%)	25 (35%)
Right sided colon	27 (35%)	20 (28%)
Rectum	23 (29%)	18 (25%)
Transverse colon	3 (4%)	8 (11%)
**Surgery for primary tumor**		
No	36 (46%)	32 (45%)
Yes	43 (54%)	39 (55%)
**Prior adjuvant chemotherapy**		
Yes	18 (23%)	10 (14%)
No	61 (77%)	61 (86%)
**Presentation at diagnosis**		
Synchronous	29 (37%)	30 (42%)
Metachronous	50 (63%)	41 (58%)
**Number of metastatic sites**		
1–2	45 (57%)	45 (63%)
3≤	34 (43%)	26 (37%)
**Sites of metastatic disease**		
Liver	55 (70%)	40 (56%)
Lung	33 (42%)	27 (38%)
Lymph nodes	41 (52%)	28 (39%)
Peritoneum	11 (14%)	28 (39%)
Bone	3 (4%)	3 (4%)
**Surgery for metastases**		
No	72 (91%)	64 (90%)
Yes	7 (9%)	7 (10%)
**RAS mutation status**		
Wild type	29 (37%)	28 (39%)
Mutant	29 (37%)	26 (37%)
Unknown	21 (27%)	17 (24%)
**BRAF mutation status**		
Wild type	43 (54%)	36 (51%)
Mutant	10 (13%)	10 (14%)
Unknown	26 (33%)	25 (35%)
**Self-reported weight-loss >5% within the last 2 months**		
Yes	55 (70%)	59 (83%)
No	24 (30%)	12 (17%)

* One missing item.

**Table 2 cancers-13-02604-t002:** Mean values of the European Organization of Research and Treatment of Cancer QLQ-C30 scores per domains by treatment group and time points. The selected domains are of special interest in older patient with mCRC. No statistically significant differences were observed between treatment arms at baseline.

	Baseline *n* = 150		9-Week Follow-Up *n* = 100		18-Week Follow-Up *n* = 60	
EORTC QLQ-C30 Domains	S1	SOx	S1	SOx	S1	SOx
	Mean (SD)	Mean (SD)	Mean (SD)	Mean (SD)	Mean (SD)	Mean (SD)
Global QoL	66.35 (22.31)	65.38 (20.92)	62.50 (20.18)	67.88 (22.01)	69.79 (23.16)	64.51 (21.38)
Physical functioning	81.60 (16.30)	77.11 (19.00)	73.65 (20.42)	77.60 (19.06)	72.32 (21.69)	77.90 (18.65)
Social functioning	83.12 (24.98)	82.39 (23.04)	82.37 (25.23)	84.03 (17.52)	83.33 (28.08)	83.95 (19.87)
Role functioning	80.38 (25.28)	77.23 (27.64)	73.08 (29.55)	75.53 (23.79)	78.28 (26.51)	80.25 (20.69)
Fatigue	32.91 (23.98)	34.98 (23.30)	40.49 (22.85)	37.27 (22.28)	38.38 (27.23)	33.33 (18.49)
Nausea and vomiting	4.85 (10.04)	6.34 (15.00)	7.69 (12.98)	11.11 (16.25)	8.59 (13.26)	7.41 (11.63)
Diarrhea	22.36 (31.45)	18.78 (25.65)	25.33 (31.99)	22.22 (26.93)	19.19 (26.39)	14.10 (21.44)
Pain	20.46 (26.55)	12.44 (19.86)	13.78 (21.82)	11.11 (20.15)	13.13 (25.94)	17.28 (24.67)
Dyspnea	16.24 (22.63)	16.43 (23.82)	25.00 (26.30)	12.50 (21.33)	20.20 (21.95)	12.82 (16.54)

**Table 3 cancers-13-02604-t003:** Mean change coefficients and between-group differences with 95% confidence intervals of the EORTC QLQ-C3 domains from baseline to 9-week and 18-week follow-ups in the two treatment arms. Anchor based clinically relevant minimally important differences (MID) in patients with colorectal cancer receiving chemotherapy according to Musoro et al. are highlighted.

EORTC QLQ-C30 Domains	Change from Baseline to 9-Week Follow-Up				Change from Baseline to 18-Week Follow-Up			
	S1	SOx	Difference		S1	SOx	Difference	
	Coefficient (95% CI)	Coefficient (95% CI)	Coefficient (95% CI)	*p*-Value	Coefficient (95% CI)	Coefficient (95% CI)	Coefficient (95% CI)	*p*-Value
**Global QoL**	−4.22 (−9.16; 0.72)	2.63 (−4.84; 10.11)	**6.85** **(−1.94; 15.65)**	0.127	0.00 (−6.16; 6.16)	−3.01 (−11.89; 5.87)	−3.01 (−13.68; 7.66)	0.58
**Physical functioning**	−7.52 (−12.23; −2.80)	−0.14 (−4.73; 4.44)	**7.37** **(0.70; 14.05)**	**0.03**	−10.44 (−16.98; −3.91)	−2.57 (−9.50; 4.36)	**7.88** **(−1.68; 17.43)**	0.106
**Social functioning**	−1.58 (−6.59; 3.43)	1.83 (−5.47; 9.13)	3.41 (−5.33; 12.16)	0.444	−2.43 (−10.12; 5.26)	0.98 (−8.85; 10.80)	3.41 (−8.93; 15.75)	0.588
**Role functioning**	−8.20 (−15.70; −0.71)	−2.86 (−10.83; 5.12)	5.35 (−5.42; 16.11)	0.33	−6.18 (−14.30; 1.94)	−1.10 (−11.10; 8.90)	5.08 (−7.96; 18.12)	0.445
**Fatigue**	7.88 (2.02; 13.73)	2.62 (−4.73; 9.96)	−5.26 (−14.81; 4.30)	0.281	8.78 (0.21; 17.36)	1.24 (−7.93; 10.40)	**−7.54** **(−20.09; 5.00)**	0.239
**Nausea and vomiting**	3.01 (−0.95; 6.97)	4.81 (0.79; 8.84)	1.80 (−3.99; 7.60)	0.542	4.40 (0.16; 8.65)	1.92 (−1.64; 5.47)	−2.48 (−8.05; 3.08)	0.382
**Diarrhea**	2.54 (−7.90; 12.98)	4.77 (−3.63; 13.17)	2.23 (−11.09; 15.54)	0.743	−2.78 (−-11.52; 5.96)	−2.91 (−11.64; 5.82)	−0.13 (−12.21; 11.95)	0.984
**Pain**	−5.72 (−12.71; 1.28)	−2.20 (−9.51; 5.12)	3.52 (−6.64; 13.67)	0.497	−2.87 (−10.10; 4.36)	4.52 (−6.72; 15.76)	7.38 (−6.24; 21.01)	0.288
**Dyspnea**	8.27 (1.70; 14.85)	−3.08 (−10.18; 4.01)	**−11.36** **(−21.18; −1.53)**	**0.024**	5.66 (−0.53; 11.84)	−1.91 (−10.35; 6.53)	−7.57 (−18.05; 2.91)	0.157

*p* < 0.05 is considered statistically significant.

**Table 4 cancers-13-02604-t004:** Risk of deterioration based on ECOG PS deterioration at 9-week and at 18-week follow-ups.

Risk of Deterioration at	9 Weeks		18 Weeks	
	OR (95% CI)	*p*-value	OR (95% CI)	*p*-Value
Main analysis	0.75 (0.40; 1.41)	0.366	0.88 (0.47; 1.64)	0.676
Sensitivity analysis A	0.96 (0.39; 2.35)	0.924	0.73 (0.32; 1.71)	0.472
Sensitivity analysis B	1.15 (0.45; 2.92)	0.771	0.83 (0.25; 2.74)	0.762

*p* < 0.05 is considered statistically significant.

**Table 5 cancers-13-02604-t005:** ECOG PS at baseline, at 9-week and at 18-week follow-ups per treatment groups indicated as counts and proportions. The sensitivity analyses with different imputation choices allowed utilization of data in incomplete observations.

	Baseline		9-Week Follow-Up		18-Week Follow-Up	
Treatment Arms	S1	SOx	S1	SOx	S1	SOx
	**Main analysis**					
PS = 0	31 (37%)	23 (30%)	19 (23%)	12 (16%)	16 (19%)	10 (13%)
PS = 1	35 (42%)	37 (48%)	28 (34%)	37 (48%)	20 (24%)	25 (33%)
PS = 2	15 (18%)	15 (20%)	8 (10%)	10 (13%)	4 (5%)	5 (7%)
Dropout/Missing	2 (2%)	2 (3%)	26 (31%)	18 (23%)	32 (38%)	28 (36%)
Death	0 (0%)	0 (0%)	2 (2%)	0 (0%)	11 (13%)	9 (12%)
	**Sensitivity analysis A**					
PS = 0	31 (38%)	23 (31%)	19 (33%)	12 (20%)	16 (31%)	10 (20%)
PS = 1	35 (43%)	37 (49%)	28 (49%)	37 (63%)	20 (39%)	25 (51%)
PS = 2	15 (19%)	15 (20%)	8 (14%)	10 (17%)	4 (8%)	5 (10%)
Death	0 (0%)	0 (0%)	2 (4%)	0 (0%)	11 (22%)	9 (18%)
	**Sensitivity analysis B**					
PS = 0	31 (38%)	23 (31%)	19 (35%)	12 (20%)	16 (40%)	10 (25%)
PS = 1	35 (43%)	37 (49%)	28 (51%)	37 (63%)	20 (50%)	25 (63%)
PS = 2	15 (19%)	15 (20%)	8 (15%)	10 (17%)	4 (10%)	5 (13%)

## Data Availability

Not applicable.

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
