# Peer review of "Quality of Life in Vulnerable Older Patients with Metastatic Colorectal Cancer Receiving Palliative Chemotherapy—The Randomized NORDIC9-Study"

_cancers, 2021, doi:10.3390/cancers13112604_

Round 1

Reviewer 1 Report

This is a profoundly serious study on the treatment of metastatic CRC in vulnerable older patients. The study is aimed to particularly evaluate the impact of treatment on patients QoL. This is a complete report, professionally written, but very dense, difficult and time consuming to read. However, the report is complete, it is in-depth description.

I will just focus on some specific points.

  • Which is the definition of “vulnerable” older patients which is practically used. Which criteria based on the “Geriatric screening tools (GST) (G-8, VES-13, Timed-Up-and-Go, Handgrip strength), Charlson Comorbidity Index and QoL completed before randomization” (Winther et al BMC Cancer 2017) are used to define?
  • I do not understand exactly the compatibility between “We hypothesized that treatment with reduced-dose SOx might result in inferior QoL and physical functioning compared to treatment with full-dose S1 monotherapy” in the introduction and “The primary endpoint of the current analyses was to test the hypothesis that treatment with reduced-dose SOx is not inferior to full-dose S1 in terms of affecting global QoL” in paragraph 2.6.1.
  • Bevacizumab is mentioned as possible treatment associated with chemo (No Cetuximab?) but is not analyzed. Why? Bevacizumab may worsen QoL.
  • It is specified that patients were not candidate to standard combination chemo: which chemo? According to which criteria?
  • The last patient was included in October 2017: this report is written 3 years later; why?
  • Finally, the results of this study are:
    1. “SOx resulted in significantly improved progression-free survival (PFS) (SOx; 6.2 vs S1; 5.3 months, hazard ratio: 0.72 (95%CI: 0.52-0.99), p=0.047). Additionally, patients treated with SOx experienced less toxicity, and fewer hospitalizations, compared to S1.” What is the impact on Overall Survival (none?)?
    2. “The between-group difference in global QoL was 6.85 (95%CI -1.94; 15.65) at 9-week in favor of SOx (SOx: 2.63 [-4.84; 10.11], S1: -4.22 [-9.16; 0.72], p=0.127). Despite this being non-significant the difference exceeded the pre-specified threshold for MID between groups. At 18-week, the difference was neither statistically significant nor clinically relevant.”
    3. Therefore PFS (one month difference) is statistically significant and global QoL at 9 months statistically significant. Is it clinically significant? My feeling is that Sox is less toxic, the most important issue, …. but what is the impact on survival and symptoms when Sox is compared to no chemo and only palliative cares?

Author Response

Dear Referee,

First, we would like to thank you for your time and effort reviewing our work. We are grateful for your excellent comments and suggestions. We answered your questions, made changes, and edits in the manuscript as you suggested, highlighted in the modified version of the manuscript. The new version is attached.

First referees report:

Which is the definition of “vulnerable” older patients, which is practically used. Which criteria based on the “Geriatric screening tools (GST) (G-8, VES-13, Timed-Up-and-Go, Handgrip strength), Charlson Comorbidity Index and QoL completed before randomization” (Winther et al BMC Cancer 2017) are used to define?

Comment from the author:

Older patients, who were not considered as candidates for full-dose combination chemotherapy (e.g.: FOLFOX, FOLFIRI, CAPEOX, SOx) were included in the NORDIC-9 study based on the treating physicians clinical judgement. The results of the geriatric screening did not influenced the randomization. We chose the widely used and validated geriatric screening tools to explore their prognostic role both in univariable and multivariable analyses. We are planning further analysis of the geriatric screening results and comparing them to well-known frailty models (Fried, Rockwood) as a part of a current phd project.  

  • I do not understand exactly the compatibility between “We hypothesized that treatment with reduced-dose SOx might result in inferior QoL and physical functioning compared to treatment with full-dose S1 monotherapy” in the introduction and “The primary endpoint of the current analyses was to test the hypothesis that treatment with reduced-dose SOx is not inferior to full-dose S1 in terms of affecting global QoL” in paragraph 2.6.1.

Comment from the author:

Given the administration of two cytotoxic agents in reduced dose we expected somewhat higher toxicities compared to full-dose single agent chemotherapy, which possibly may affect global QoL negatively and result in inferior QoL outcomes. However, patients receiving reduced-dose combination treatment experienced less toxicities, therefore, testing for non-inferiority between the two arms seemed appropriate.

  • Bevacizumab is mentioned as possible treatment associated with chemo (No Cetuximab?) but is not analyzed. Why? Bevacizumab may worsen QoL.

Comment from the author:

In patients with RAS and BRAF wild-type colorectal cancer, the first-line treatment usually consists of a cytotoxic doublet (FOLFOX; FOLFIRI) combined with an EGFR inhibitor, however, in older patients, not candidate to intensive chemotherapy, or presented with a metastatic disease, unlikely become resectable, where the goal of the treatment is to avoid progression, capecitabine in combination with bevacizumab is a well-accepted option based on the AVEX-study. In later lines, these patients might receive reduced dose doublet with EGFRi, or irinotecan + EGFRi, or EGFRi monotherapy. In our study, we chose three-weekly regimens with peroral S1. With regards to international guidelines (ESMO) the combination of EGFRi with three-weekly regimens including capecitabine, S1 are not recommended and should be avoided. Thus, treatment with EGFRi was not a part of the design of the NORDIC9-study.

In the NORDIC9-study 28% of the patients received bevacizumab in addition to chemotherapy. In most Nordic countries bevacizumab was not standard of care at the time where the NORDIC9-study was design, because no first line randomised trial had shown survival benefit with combination chemotherapy. Thus, based on the positive data from the AVEX-trial, the use of bevacizumab was optional. Reviewing available data in the literature, and considering our experience, bevacizumab monotherapy is unlikely affecting global QoL, hence, we did not include bevacizumab in QoL analyses.

  • It is specified that patients were not candidate to standard combination chemo: which chemo? According to which criteria?

Comment from the author:

Full-dose combination chemotherapy regimens containing a backbone of 5-flouroruracil (also including peroral agents such as Capecitabine and S-1), combined with either irinotecan or oxaliplatin (FOLFIRI, FOLFOX, CAPEOX (CAPEIRI – less used, SOx, IRIS), are standard options in fit or younger patients with metastatic colorectal cancer. Evidence-based recommendations can be found in e.g.: ESMO, ASCO, and NCCN guidelines.

  • The last patient was included in October 2017: this report is written 3 years later; why?

Comment from the author:

The NORDIC9-study is an investigator-initiated project, further pre-planned analyses including quality of life- data are a part of a new PhD project, and we have spent almost 2-years getting the project funded.

  • Finally, the results of this study are:

“SOx resulted in significantly improved progression-free survival (PFS) (SOx; 6.2 vs S1; 5.3 months, hazard ratio: 0.72 (95%CI: 0.52-0.99), p=0.047). Additionally, patients treated with SOx experienced less toxicity, and fewer hospitalizations, compared to S1.” What is the impact on Overall Survival (none?)?

Comment from the author:

The treatment with SOx prolonged the OS with 3 months (14.5 vs 11.5), however, this was not statistically significant (Winther et al, Lancet GH, 2019). In our opinion, we need further investigation in this older, vulnerable population, and we cannot conclude yet, that this strategy would not result in OS benefit.  

  • “The between-group difference in global QoL was 6.85 (95%CI -1.94; 15.65) at 9-week in favor of SOx (SOx: 2.63 [-4.84; 10.11], S1: -4.22 [-9.16; 0.72], p=0.127). Despite this being non-significant the difference exceeded the pre-specified threshold for MID between groups. At 18-week, the difference was neither statistically significant nor clinically relevant.

”Therefore PFS (one month difference) is statistically significant and global QoL at 9 months statistically significant. Is it clinically significant? My feeling is that Sox is less toxic, the most important issue, …. but what is the impact on survival and symptoms when Sox is compared to no chemo and only palliative cares?

Comment from the author:

In older, vulnerable patients with limited life expectancies, SOx has shown superior efficacy, improved PFS, and better QoL, thus, in our opinion these results may be considered clinically meaningful and significant.

Our study was not designed to comparing reduced-dose SOx, or full-dose S1 to best supportive care. However, any treatment resulting in symptom-relief and maintained/improved QoL might be considered meaningful in this group of patients often prioritizing QoL and other patient-centered endpoints over improved survival. Indeed, we can understand your concerns, and agree that this topic needs more attention and further investigation.

We believe that the quality of our manuscript is improved, and it is now worth being reconsidered for publication.

 Sincerely yours,

Gabor Liposits

Reviewer 2 Report

This study is a subanalysis of the randomized phase II NORDIC9 study (PMID: 30852136). Their previous paper reporting the primary endpoint showed the higher efficacy and less toxicity of reduced SOX strategy than full-dose monotherapy with S-1. In the present study, the authors have reported another clinical meaning of reduced SOX strategy in vulnerable patients with metastatic colorectal cancer: non-inferiority of QOL compared to full dose S1 monotherapy. It is well written. The following minor questions are posed to the authors

  • In Table 3, the authors use “S1 versus Sox” at the 9-week results but use “monotherapy versus Reduced dose doublet” at the 18-week results. Should use the same terms.  
  • There are many results shown. It seems that the 3-7. Baysian analyses do not support their main findings well. It may be better to take out this section or put as a supplementary file.

Author Response

Dear Referee,

First, we would like to thank you for your time and effort reviewing our work. We are grateful for your excellent comments and suggestions. We answered your questions, made changes, and edits in the manuscript as you suggested, highlighted in the modified version of the manuscript. The new version is attached.

Second referees report:

  • In Table 3, the authors use “S1 versus Sox” at the 9-week results but use “monotherapy versus Reduced dose doublet” at the 18-week results. Should use the same terms.  

Comment from the author:

Thank you, we made the desired change in Table 3, using uniform terms (S1 and SOx).

  • There are many results shown. It seems that the 3-7. Baysian analyses do not support their main findings well. It may be better to take out this section or put as a supplementary file.

Comment from the author:

We will suggest the editor to move Table 6 and Figure 3 to a supplemental file. 

We believe that the quality of our manuscript is improved, and it is now worth being reconsidered for publication.

 Sincerely yours,

Gabor Liposits